# Retrospective challenges to pre-exposure prophylaxis (PrEP) use among people living with HIV—A qualitative analysis using the COM-B framework

Carina Hörst[1]*, Hannah Kitt[1], Helen Corkin[1], Dolores Mullen[1], Ammi Shah[1], Adamma Aghaizu[1], Clare Humphreys[2], Tamara Djuretic[1]

1 Blood Safety, Hepatitis, STI & HIV Division (BSHSH), UK Health Security Agency, London, United Kingdom, 2 UK Health Security Agency South East, Chilton, United Kingdom

☯ These authors contributed equally to this work
* carina.hoerst@ukhsa.gov.uk

## Abstract

Biomedical interventions to prevent transmission of Human Immunodeficiency Virus (HIV), such as pre-exposure prophylaxis (PrEP), are available in England for free at specialised sexual health services (SSHS). Yet the number of new HIV diagnoses made in England among people exposed to HIV through heterosexual sex and among people of Black ethnicity are increasing. Published research discusses the barriers and facilitators to PrEP among people not living with HIV. We add to this literature by qualitatively exploring the retrospective barriers and facilitators to PrEP use and HIV prevention more broadly among people who are living with HIV now. We interviewed 26 participants between March 2021 and July 2022 from across England with varying backgrounds. We used the COM-B model to systematically extract areas that can inform behaviour change. Capability barriers included gaps in HIV and PrEP knowledge and mental health issues. Opportunity barriers included not being identified as having a PrEP need by healthcare professionals, which then hindered participants to learn more about PrEP, communication gaps with sexual partners and perceptions of (in)accessibility of PrEP. Motivational barriers included a perception of low HIV risk and HIV/ PrEP self-relevance, PrEP stigma, having reservations about taking medications, and perceiving HIV campaigns as too selective regarding the populations they target. Knowledge about PrEP constituted a capability facilitator; and wanting to stay safe and take control of one's health was a motivational facilitator. Our findings mirror those identified in research conducted with people not living with HIV, but we believe that particularly strong narratives for HIV prevention can arise from learning from people living with HIV. We suggest that structural changes are required to achieve a shift in how sex, HIV and PrEP are discussed societally, allowing for changes in individual and interpersonal behaviours and a sustained decrease in HIV transmission among all groups.

**Data availability statement:** The data that support the findings of this study have been assessed by the UK Health Security Agency Office for Data Acquisition and Release as having sensitive personal information and are therefore not publicly available to protect participant privacy. However, some summaries of the data may be available upon reasonable request from the UKHSA. Requests can be directed to DataAccess@ukhsa.gov.uk.

**Funding:** The author(s) received no specific funding for this work.

**Competing interests:** The authors have declared that no competing interests exist.

## Introduction

Combination HIV Prevention, proposed by UNAIDS in 2009 [1], suggests that effective HIV prevention requires the provision of medical treatment, individual or community preventative behaviour, as well as addressing and reducing stigma, discrimination, and gender inequality. Pre-exposure prophylaxis (PrEP) is one of the key HIV biomedical preventative methods. PrEP contains tenofovir with emtricitabine and can be highly effective in preventing HIV if taken as one daily oral tablet or on the days before and after sex ('event-based'). In 2015, the World Health Organization (WHO) recommended that anyone at risk of acquiring HIV should take PrEP [2]. In the UK in 2016, the predecessor of the UK Health Security Agency (UKHSA), Public Health England, and the National Health Service (NHS) England launched the PrEP Impact Trial [3]. The trial was intended to assess the costs associated with providing PrEP through specialised sexual health services (SSHS) (which are sexual health-specific, confidential and free clinics), determine the population's need for PrEP, the length of that need, the rate of PrEP uptake, and the duration of use among sexual health services (SHS) attendees in England. The trial was conducted in 157 SHSS across England between 13th October 2017 and 12th July 2020. Clinicians assessed attendees who were not living with HIV for their risk of acquiring HIV to identify eligibility to participate. In total, 24,268 participants were enrolled in the PrEP Impact Trial, the majority of whom were gay and bisexual and other men who have sex with men (GBMSM) (20,403), of White ethnicity (16,111), and UK born (13,017). PrEP became freely available in all SSHS in England in October 2020 [4]. Tenofovir disoproxil and emtricitabine (Truvada) is now available on the NHS. However, in the UK, the PrEP mostly used is generic PrEP – a first-line PrEP which can be used for event-based sex. An alternative is tenofovir alafenamide and emtricitabine (Descovy) which is prescribed to people with kidney issues, osteoporosis or under 18 years [5–7]. In 2024, the UK Medicines and Healthcare products Regulatory Agency approved of the injection of cabotegravir as an additional means of taking PrEP [8], which is yet to be made widely available.

Interventions aimed at preventing HIV acquisition are only successful if their delivery is comprehensive and equitable. Whilst HIV prevention strategies in England resulted in a decline in new diagnoses in 2023 among men exposed through sex with men by 35% from 2019 [9], the number of HIV diagnoses first made in England continues to rise among other demographic groups. In England, from 2022 to 2023, these increased by 32% among men and women exposed to HIV via heterosexual sex, accounting for almost half of all new cases (49%) [9]. HIV diagnoses rose by 36% for men and by 30% for women during this period. Among GBMSM, there was a 3% increase in new diagnoses among all White men, whilst the increase for all other ethnic groups combined was more than double (7%). The greatest increase was observed among GBMSM of Black ethnicity (42%).The increase of new diagnoses among White people exposed to HIV through heterosexual sex was 3%, whilst the increase among heterosexual people of all other ethnic groups combined was 45%. The strongest increase was observed for heterosexual people of Black African ethnicity (64%; however, it was noted that this number could potentially overestimated due

to a possible mis-categorisation of people as new diagnoses when they were previously diagnosed with HIV and receiving care abroad). Whilst the reasons for this increase are multifactorial, it suggests that HIV prevention is not accessible to or utilised by people of different sexual orientations and ethnicities equally.

The barriers (and facilitators) to HIV prevention, and PrEP specifically, among different populations at risk of HIV are well researched. A systematic review of quantitative, qualitative and mixed-methods studies (predominantly conducted in the U.S. and African countries) identified individual, social and interpersonal and structural barriers to the uptake of PrEP among people (including GBMSM, young adults, trans and ciswomen, sex workers, Black and ethnic minoritised people) who were currently or previously taking PrEP [10]. Individual barriers related to, for example, concerns about medical side effects, difficulty adhering to daily tablets, perception of low individual risk for HIV, mental health problems and unplanned sexual encounters (when participants were using "event-based" PrEP). Social and interpersonal barriers included PrEP stigma (e.g., fearing that PrEP uptake would be associated with certain stigmatised sexual behaviours and identities), and a lack of partner support (e.g., fearing to be viewed as sexually not monogamous). Structural barriers included limited access to PrEP, long waiting times, negative or insensitive attitudes from healthcare workers and high financial costs. Insights from the African continent reveal the influence of cultural factors such as family pressures impacting PrEP uptake [11], or negative parental influence, for example, among young people in Uganda, Zimbabwe or South Africa where sex at young age and before marriage is culturally disapproved of [12]). Evidence suggests that cultural and religious factors that can shape perceptions and willingness to use PrEP (e.g., discomfort discussing sex and sexual health, misconceptions that PrEP is a treatment for HIV, or concerns about sexual fidelity) often persist when individuals migrate to other places or countries, as seen among communities in the African diaspora settling the U.S. or the UK [13]. Therefore, culturally-competent healthcare provider–patient communication has been shown critical in empowering women of Black African and Caribbean descent living in the UK to use PrEP [14]. A systematic review from the UK synthesised studies that employed quantitative, qualitative, and mixed-methods approaches, which focused on people not living with HIV – primarily GBMSM, including a small number of Black participants, as well as women (including trans women), Black and ethnic minority groups, and people who inject drugs – who had accessed SSHS [15]. Identified barriers and facilitators to the uptake of PrEP greatly overlapped with the aforementioned. However, the authors used a behaviour change framework – the Motivational PrEP Care Continuum [16] specifying five steps from 'pre-contemplation' to' PrEP maintenance' stage – which provided a more systematic overview of where the barriers and facilitators are located in individuals' PrEP journey. Whilst lack of PrEP knowledge and self-perception of HIV risk were, for example, categorised in the earlier stages on the continuum, perceptions of PrEP stigma, eligibility criteria and access fell into the later stages. A follow up qualitative study, conducted by the same first author [17] explored the barriers and facilitators to PrEP uptake specifically among Black African, Black British and Black Caribbean women in England. They employed the widely used COM-B model [18] – assigning barriers and facilitators to the model's domains of individual capability and motivation, as well as social and environmental opportunities. The most significant individual-level barriers to PrEP uptake were gaps in knowledge, largely stemming from limited information provided to Black women. At the provider level, restrictive policies around eligibility, exclusive provision through sexual health services, and priorisation practices were identified as key barriers. Conversely, enhanced PrEP education, broader policy access, community engagement, and trusted messengers promoting bodily autonomy and empowerment were identified as strong facilitators.

## This study

Barriers and facilitators to PrEP use are well researched among people not living with HIV [10–15,17,19]. But, to the best of our knowledge, examining the perceptions of people who are now living with HIV on their views and use of PrEP prior to diagnosis is not commonly integrated into this work. This has been identified as a research gap [20]. Whilst for people not living with HIV, acquiring the virus may be hypothetical and possibly perceived as psychologically distant [21], for people living with HIV, it has become part of their reality. They therefore may be able to retrospectively identify the most

critical barriers to PrEP which could then be removed for others. We believe that these perspectives are currently missing from the literature on the barriers and facilitators to PrEP, and that they can complement research conducted with people not living with HIV. Reflections on past behaviours, experiences, and actions (or the lack thereof) may identify which barriers were particularly influential in decision making on preventive measures, including PrEP. This can therefore potentially highlight obstacles that may be underestimated by people not living with HIV and produce stronger narratives to inform HIV prevention work.

The UK Government has committed to WHO's goal to achieve zero new HIV transmissions by 2030, and a reduction in new HIV diagnoses by 80% from the 2019 baseline year to 2025 [22]. Whilst it is unlikely that this interim goal will be achieved [23], efforts need to be enhanced to achieve zero HIV transmission including a strong focus on HIV prevention. The commitment included a pledge to making access to HIV prevention, including PrEP, equitable. In 2018, UKHSA started enhanced surveillance to better understand why HIV transmission continues despite available HIV prevention (e.g., PrEP, post-exposure prophylaxis (PEP), condoms, early HIV testing) and facilitation of prompt diagnosis (free community-based HIV testing, home testing, HIV testing in a range of healthcare settings). The surveillance qualitatively explored the barriers that people experienced to using HIV prevention methods, the context in which people acquire HIV and their knowledge of and attitudes towards available prevention options. This initiative has been referred to as 'SHARE' – 'Surveillance of HIV Acquired Recently: Enhanced' [24], and expanded upon existing questionnaires for clinicians and patients, with the implementation of semi-structured interviews with people living with HIV. Using information gathered in these interviews, we aimed to produce a comprehensive account of the experiential journey of participants prior to HIV diagnosis to provide insights into their perceptions, knowledge, and beliefs about HIV, their lived experiences, and the challenges and enablers for their use of HIV prevention, access to, and engagement with health services. We applied the COM-B model [18] to the data to understand the barriers and facilitators to consideration of using PrEP, as well as overarching factors constituting barriers and facilitators to HIV prevention more broadly.

## Methodology

### Design and theoretical framework

We conducted a thematical framework analysis of semi-structured interview data. We employed the widely used COM-B model [18] which is linked to intervention frameworks, such as the Behaviour Change Wheel (BCW), and can, therefore, derive information about the most suitable interventions to improve corresponding behaviour. We undertook a qualitative thematic analysis with a hybrid approach to coding (deductive coding by assigning data to the COM-B domains; and inductive coding by deriving themes within the domains) to understand the patterns (barriers and facilitators) of engagement or disengagement with PrEP among participants. Insights are retrospective since all participants were now living with HIV, and accounts of PrEP use facilitators are predominantly hypothetical since most participants had not used PrEP. However, since most participants had at least thought about doing so at the time, we were able to explore the barriers to PrEP, as well as what facilitated the *consideration* of using PrEP. Participants' perceptions and experiences are subjective and may not always reflect objective opportunities or health advise, but we treat these as actually perceived and/ or experienced by participants to inform and improve future HIV prevention work. We therefore take a pragmatist approach to the data [25].

The COM-B model is widely used in public health research and describes how Capability, Opportunity, and Motivation influence Behaviour [18]. Capability refers to psychological (i.e., knowledge, memory, attention and decision-making processes, skills and behavioural regulation), and physiological functioning (i.e., a person's physique and musculoskeletal functioning); opportunity refers to physical (i.e., inanimate parts of the environment, a system or time) and social opportunities (i.e., other people and organisations, including culture and social norms); and motivation comprises reflective (i.e., conscious thoughts such as plans and evaluations) and automatic motivation (i.e., habitual, and instinctive motivations, including desires). The model is linked to intervention frameworks, such as the Behaviour Change Wheel (BCW), which

can inform behaviour change interventions. In this research, we explored whether participants had the capability, opportunity, and motivation to engage with HIV prevention, specifically PrEP.

## Ethical considerations

This study was reviewed and approved by the UKHSA Research Ethics and Governance Group (REGG), no. NR0260. The study has been classified as enhanced surveillance, undertaken as part of managing diagnoses, identifying trends, controlling, preventing, monitoring, and managing communicable diseases and other risks to public health in the population. Therefore NHS REC review and approval under the Caldicott principles [26] were not required. During recruitment, the UKHSA research team avoided recruiting people who had just received their HIV diagnosis to allow them time to adjust first. To ensure questions were sensitive, respectful and appropriate, the topic guide was developed in conjunction with patient group representatives (UK-CAB). Interviewers were recruited who were experienced with and culturally competent with HIV. Participants were informed that they could end the interview at any time and were not obligated to answer any questions they felt uncomfortable with. Participants were also given the option to decide the time of the interview and to continue it at a later time if they wished to pause. After the interview, the interviewer and participant had a post-interview debrief which included signposting participants to general organisations or helplines and tailored support for specific issues raised. They were all in contact with HIV care services who had referred them for the interviews.

## Sampling strategy

A purposive sampling approach was adopted. Participants needed to be 18 years or older and be receiving care at an HIV clinic in England. Participants were eligible if they had evidence of recent HIV acquisition (i.e., a negative HIV test in the previous two years, a positive Recent Infection Testing Algorithm (RITA) test [27], evolving serology, e.g., p24 antigen positive and antibody negative, or evidence of seroconversion illness)). UKHSA's HIV surveillance team identified people within the HIV and AIDS Reporting System (HARS) and HIV & AIDS New Diagnoses Database (HANDD) who fit the eligibility criteria and notified clinicians on which patients were eligible for invitation to participate. Clinicians were asked to describe details of the study to these participants and explain the patient information leaflet and consent form, to ensure potential participants were suitably informed. Where there was interest in taking part, HIV clinicians obtained consent to pass on their preferred contact method and details to UKHSA. For patients who agreed to take part, clinicians recorded the preferred contact details for the patient in the recruitment form. The clinician had to indicate on the recruitment form that verbal consent had been given before being able to pass on patient's contact details to UKHSA. The consent form was for patients to keep as a reminder of what they have consented to. The recruitment information sheets contained unique SHARE qualitative interview IDs to avoid collecting any personally identifiable information. Information about minimal demographic data (e.g., age group, ethnicity, country of birth, sexual orientation, date of HIV diagnosis) was obtained as part of the recruitment process (after consent had been given) or during the interviews.

## Participants

Table 1 describes the main demographic characteristic of participants. Additionally, two had a history of injection drug use, one had a history of imprisonment, and another had been a sex worker. The majority of participants were between 25 and 49 years which is representative of the age range with the highest number of new HIV diagnosis in England (66.3% in 2023 and 68.1% in 2024).

## Data collection

Two white British women (of whom HC is a co-author) with experience in qualitative interviewing and with a professional background in sexual health contacted prospective participants to arrange the interviews and provide an opportunity to ask

**Table 1. Participant demographics and participation.**

| Participation | |
| --- | --- |
| No. of participants approached | 76 |
| No. of participants consented | 45 |
| No. of participants interviewed | 26 |
| Response rate | 34% |
| **Gender identity** | |
| Men | 21 |
| Women | 5 |
| **Sexual orientation, gender and exposure** | |
| Gay – men (HIV acquired through sex with men) | 20 |
| Heterosexual – men (HIV acquired through sex with women) | 1 |
| Heterosexual – women (HIV acquired through sex with men) | 5 |
| **Ethnic group** | |
| White British | 14 |
| White other | 4 |
| Black African | 4 |
| Black Caribbean | 1 |
| Asian | 1 |
| Mixed | 2 |
| **Age group (years)** | |
| 18–24 | 2 |
| 25–34 | 10 |
| 35–49 | 11 |
| 50–64 | 3 |
| 65 and over | 0 |
| **Employment status** | |
| Employed | 16 |
| Unemployed | 5 |
| Other (in training, studying) | 2 |
| No information provided | 3 |
| **Place of birth** | |
| Europe | 22 |
| Born in the UK | 18 |
| Africa | 4 |
| **Place of residency** | |
| London | 11 |
| Outside of London | 15 |

questions prior to the interview. Interviewers introduced themselves, their connection to the NHS medics the participants were being treated by, and their role in the interview at the beginning of it in all cases. When talking to gay men, one of the interviewers who was gay herself, shared this personal background with the interviewees to create common reference points and cultural understandings. This facilitated the conversation with these participants. Where interviewees were women, explicit gender commonalities also helped the discussion. However, in some cases, revealing differences (e.g., not being from the same country or community) helped with underlining confidentiality and absence of stigma. Interviewers, however, deemed familiarity and comfort level with HIV and not holding stigmatising views as most important in building rapport with participants.

The intention was to interview up to 50 people living with HIV to allow representation of people from varying demographic and sexual risk groups (including people who inject drugs, sex workers, people with experience of incarceration) and to subsequently review if saturation had been reached. Table 1 provides an overview of the recruitment success. The recruitment started March 1st, 2021. Several attempts were made to contact eligible participants who had given their consent, yet many participants had forgotten they had agreed to participate at the time they were contacted by the interviewers, and many withdrew from recruitment at this stage. Some were unable to confirm a suitable time for an interview or had changed their mind. In total, twenty-six people were interviewed between March 25th, 2021, and July 6th, 2022. Halting recruitment was consequently determined by logistical reasons rather than data saturation.

All interviews took place remotely via a video or telephone call and lasted between 30 and 190 minutes and were recorded using a dictaphone. The interviewers opted for phone calls as using MS Teams was seen as a barrier due to its predominantly professional use. Most participants were by themselves and somewhere private when they took part which allowed talking about sensitive topics. Interviews were conducted using a topic guide (S1 Topic guide in S1 Text), co-produced by the UKHSA HIV team, interviewers and stakeholders including HIV patient representatives from the UK Community Advisory Board (UKCAB), other non-governmental organisations (NGOs) such as the Sophia Forum, Prepster and the African Health Policy Network, as well as the Rapid Research, Evaluation and Appraisal Lab (RREAL) team at University College London (UCL). SHARE was originally a questionnaire (see above 'This study') and interview questions were developed from it. Interview questions were also piloted and data from 6 participants were included in the data analysis. It included questions on demographics, reasons for testing, testing history, perceived and actual HIV risk, HIV knowledge and attitudes, experiences of HIV prevention and sexual health services and sexual health promotion campaigns. The topic guide was revised regularly based on the information collected.

The interviewers obtained and recorded verbal consent prior to the interview. Participants were remunerated for their time with a £25 gift voucher as outlined in Public Health England's payment for public involvement guidance at the time [28].

Participant contact details were kept separately from interview notes and recordings and names were deleted from transcripts. Interview recordings and in part temporary deidentified field notes (used during interviews as prompts and reflections) were stored on secure UKHSA drives and deleted from any local devices immediate after upload. All members of the UKHSA research team involved in SHARE had training in handling sensitive data according to the Caldicott principles [26].

## Data analysis

The initial approach to the data analysis followed a framework analysis approach [29], by members of the research team who had initiated the interview study. These had specialist HIV knowledge and coded the data deductively according to the topics listed in the topic guide and by listening to the audio recordings of the interviews. The framework captured general observations to which the interviewers contributed. Transcripts of the interviews were subsequently generated in NVivo 14 [30] – a software to transcribe and analyse qualitative data. For the purpose of our study focussing on the challenges associated with using PrEP, this data was re-analysed. We maintained the framework but restructured it to reflect barriers or facilitators and the domains of the COM-B model in reference to PrEP and overarching factors (see below) instead (S2 Codebook in S2 Text). We then categorized the data deductively as barriers/ facilitators and to the domains each based on best fit (i.e., no data overlapped between domains). The Theoretical Domains Framework (which extends and elaborates on the COM-B domains in more depth [31]) aided in the decision-making process. The interviews explored a wide range of direct and indirect factors that influenced PrEP use. Indirect factors were defined as 'overarching factors' which concerned HIV prevention more broadly (e.g., condom use). Within each COM-B domain, codes were generated inductively and grouped together based on similarity to generate themes of barriers and facilitators to considerations of the uptake of PrEP specifically and HIV prevention generally. We analysed the data in Excel [32] since the original framework analysis had been generated and analysed in Excel, too.

We acknowledge that background factors may have influenced the data. Although none of our participants took part in the PrEP Impact Trial [3], they were recruited during or after its conclusion, so the identified barriers and facilitators for PrEP use vary due to the change in PrEP availability and policies. The interviews also overlapped with the COVID19 pandemic. Access to sexual healthcare and consultation were impacted by this, which may have further negatively influenced the uptake of PrEP [33]. Throughout the analysis, we provide the diagnosis date, where relevant, to provide context on these potential influences.

At the time when the current authors took over from the research team who had initiated the interview study, the final sample was predetermined (see Data collection for the reasons on that). Therefore, we cannot conclusively determine whether thematic saturation has been achieved. However, although some subpopulations were more dominantly presented (i.e., GBMSM) than others (e.g., women), all of our participants shared an experience – a recent HIV diagnosis. This was at the heart of our research question and makes our sample specific enough to provide sufficient insight [34]. The overall heterogenous composition regarding gender, ethnicity, socio-economic class and age provides internal variation which can be beneficial to explore how the shared experience possibly differently manifests. Though we cannot conclude it, we are nonetheless confident that we have identified sufficient thematical insight to provide insights into the barriers and facilitators to consideration of using PrEP as well as overarching factors constituting barriers and facilitators to HIV prevention more broadly among people who are now living with HIV.

### Researcher characteristics and reflexivity

The study team drew from a variety of disciplines and methodological backgrounds. CHö and DM worked as Behavioural Scientists (CHö has a PhD in Social Psychology, DM has a Master's degree in Osteopathy), HK, AS and AA as HIV/ STI Surveillance and Prevention Scientists (HK and AS have a Master's degree in the Control of Infectious Diseases, and AA has a PhD in surveillance and epidemiology of HIV), HC was a Sexual Health and HIV Lead and holds a Bachelor degree in Social Anthropology and a Masters in Global Public Health, and TD (FFPH PhD) and CHu a Consultant Epidemiologist (FFPH, MSc Demography and Health).

The two interviewers had previously worked as sexual health advisors and were familiar with working on HIV and did not hold stigmatising attitudes. CHö, HK, and DM analysed the data. HK had prior working experience with HIV as an HIV/ STI Surveillance and Prevention Scientist, and DM had previously worked as an Information Analyst with a focus on STIs. CHö was new to the topic. Both interviewers and one of the analysts (CHö) had prior experience with conducting qualitative research.

The interdisciplinary team composition allowed for open discussion about the interpretation of the data, as well as mutual learning by frequently discussing questions about the methodology, and the subject of HIV and PrEP among all authors. To enhance the reliability of the coding process, codes were first generated by CHö and then cross-checked by HK. The analysts discussed discrepancies and reached an agreement on final codes. Themes were also discussed among all authors and with the study's Steering Group which included experts of all mentioned disciplines as well as HIV practitioners.

## Results

Table 2 provides an overview of barriers and facilitators by COM-B domain regarding overarching factors relating to HIV prevention and PrEP use specifically. In the following, we will first present and discuss the barriers and then the facilitators.

### Barriers

**Capability. Gaps in HIV knowledge:** Some participants presented gaps in knowledge about HIV, or accurately identifying HIV risk or prevention. We identified these gaps across a range of sexual orientations, ages, and ethnicities:

**Table 2. Barriers and facilitators to uptake and consideration of general HIV prevention and PrEP by COM-B domain.**

| COM-B domain | Barriers/facilitators | Theme (number of participants mentioning underlying accounts) |
|---|---|---|
| Capability | Barriers | • Gaps in HIV knowledge (6) *<br>• Gaps in PrEP knowledge (11) Δ<br>• Mental health (7) * |
| | Facilitators | • Gaining knowledge about PrEP (10) Δ |
| Opportunity | Barriers | • Unmet needs to learn more about HIV (13) *<br>• Unmet needs in healthcare settings to learn more about or to receive PrEP (12) Δ<br>• Communication gaps about sexual health in sexual relationships (9) *<br>• PrEP is inaccessible (7) Δ |
| | Facilitators | --- |
| Motivation | Barriers | • Low self-relevance for HIV (14) *<br>• Low self-relevance for PrEP (7) Δ<br>• PrEP scepticism and reluctance (7) Δ<br>• PrEP stigma (4) Δ<br>• Existing HIV campaigns are too selective (2) * |
| | Facilitators | • Wanting to stay safe and take control of one's health (5) Δ |

Please note:

∗ Indicates themes relating 'overarching factors' to HIV prevention generally.

Δ Indicates themes that concern PrEP specifically.

"I just knew that in the 1980's and 90's, [HIV] killed a lot of people in Africa, that was it and when you got it, you're dead" (Heterosexual man, Asian ethnicity, 35 to 49 years)

"I had just a basic understanding of what [HIV] is and how medicine of advanced so far." (Heterosexual woman, Black African ethnicity, 35 to 49 years)

"I think I did [know about HIV prevention tools], but I would say, but limited" (Gay man, White British ethnicity, 50 to 64 years)

**Gaps in PrEP knowledge:** Some participants showed gaps in knowledge about PrEP specifically. The gaps concerned having no or insufficient knowledge about PrEP generally, not knowing the difference between *pre*-exposure (PrEP) and *post*-exposure prophylaxis (PEP; an oral tablet to take within 24–72 hours after possible exposure) as well as that PrEP could be taken 'event-based' rather than daily, or how to access PrEP:

"So, I knew of, I know people some people were on [PrEP], but I wouldn't have had a clue how to get it. You know, how it starts, how, you know. […] Whether [PrEP is available from the] NHS, whether I would have been old enough to have it. I wouldn't. I don't know". (Gay man, Mixed ethnicity, 15 to 24 years, diagnosed with HIV more than 6 months after the PrEP Impact Trial had ended)

We believe that having gaps in PrEP knowledge would be linked to having gaps in general HIV knowledge. Whilst we only identified one person who explicitly expressed gaps in both, most participants either mentioned a lack of knowledge about HIV or about PrEP specifically. This could be because general HIV and PrEP specific knowledge were not explored consistently across all interviews. The initial focus of the interviews was not exclusively on PrEP uptake, and PrEP may have been less relevant for participants post-diagnosis so that it was not consistently discussed and explored compared to other topics.

**Mental health:** Some participants reported that their mental health, at least at times, negatively impacted their perceptions of risk. The reasons varied, but the impact on participants was commonly to feel ambivalent regarding their health.

Alongside this, participants reported that despite offers of some social support (e.g., outreach by health care providers), they were not able at the time to prioritise their sexual health. Participants were sometimes unable to recognise or engage in prevention (mostly condom use). Mental health issues can reduce cognitive abilities, such as decision-making, problem-solving or focus, as well as beliefs about capabilities (i.e., motivations), for example, self-esteem, which may reduce someone's capacity to engage with health care, making this a psychological capability barrier.

> "[…] I think I was a lost cause really; I think they [healthcare providers] did, they did pretty much everything they could to help me. I mean, I can remember one occasion it was literally like 'What are you doing? This is not going to end well, you're throwing your life away'. Um, and yeah, I mean, that was that, yeah, um genuine, some kind of human kind of get a real sense of compassion and I was touched by it, you know? Yeah, I am afraid it was a waste of their time unfortunately, but it was there. [...] Yeah, I think that I engaged with some of the help and support, you know, but I think drifted further out and I think I felt a bit beyond it really." (Gay man, White British ethnicity, 35 to 49 years).

We also identified that among a few participants, the relevance of HIV – and relatedly decision-making processes – seemed to be influenced by participants' life circumstances, for example, by taking less care of their health at the time of relationships ending and turning to risky coping mechanisms (e.g., unprotected sex), due to struggling with emotional and mental health:

> "I was in a long-term relationship until [*date omitted*]. So sometime before that ended, I got involved in the party scene and it's a slow descent really from kind of living a fairly stable sort of relationship of coupledom to going to be out, and out feral I suppose." (Heterosexual man, Black African ethnicity, 50 to 64 years).

**Opportunity. Unmet needs to learn more about HIV:** Participants described several different situations in which they had unmet needs to learn more about HIV. These unmet needs are therefore strongly related to the above-mentioned gaps in HIV knowledge, though here the focus lies on a lack of social support or opportunities to *gain* this knowledge. For example, some participants mentioned that they thought there that was a lack of HIV campaigns and sex education. Yet accounts of the awareness of campaigns (including the lack thereof) in clinics and everyday life were commonly only made when prompted by the interviewer. We believe that the fact that participants did not proactively mention these could be symptomatic of the lack of awareness outlined above, as well as of ineffective campaigns (or the potential ineffectiveness of campaigns more generally) that miss reaching those who could benefit from them the most.

Some participants attributed their lack of social support to insufficient relationships and sex education at school, for example, by having been offered no discussion on gay sexual health and relationships, learning that prevention was stigmatised, or that HIV was absent from sex education altogether. It was also noted that due to less face-to-face clinic appointments at the time of the Covid-19 pandemic, opportunities for conversations with health care professionals about HIV prevention options were lacking. The barrier most often mentioned, was a lack of interpersonal support in learning more about relevant HIV prevention:

> "I was still, you know, I was literally on my own. I didn't have any other gay friends […] So, it's quite hard to find the information out there. 'You've got to go and look for yourself." (Gay man, White British ethnicity, 25 to 34 years)

**Unmet needs in healthcare settings to learn more about and to receive PrEP:** Like the overarching factor of missing opportunities to learn more about HIV, some participants explained they missed discussing and learning about PrEP at SHSS or from staff at general practices and general practitioners. Specifically, these included not being identified as having a PrEP need by a healthcare professional, for example, because participants were understood to be in a sexually

monogamous relationship or to be using condoms consistently. This was the case even with participants with a history of other diagnosed STIs:

> "Not much [talk about PrEP in clinics]. I didn't, I think I could have done with a bit more prompting. [...] Like, you know, I used to go for sexual health appointments, I've had partners with syphilis in the past, I've had chlamydia, gonorrhoea several times and I think I would have liked somebody to say 'Hey, you know, PrEP works and if you ever have, even if it's just one on an occasional basis, have unprotected sex, please really think about taking it. This is information and this is how it works and so on". (Gay man, White Other ethnicity, 35 to 44 years, diagnosed with HIV during the time of the PrEP Impact Trial, but before 2020)

On one occasion where a participant had actually used PrEP, he reported that a healthcare professional incorrectly assessed him as low risk for HIV due to his long-term monogamous relationship, and subsequently advised him to stop taking PrEP, but did not recommend an HIV test for his partner, from whom the participant believes he later acquired HIV. This stresses the need to screen sexual partners, too:

> "[…] because we're monogamous, and I spoke to the sexual health clinic, funny enough, and said 'Look, I'm in a relationship, we're monogamous, we're not actively, you know, sexual in the sense that we want to be promiscuous with those people and so forth. And we don't engage in recreational drugs to engage in activities that are at risk. [...]. They [staff at SHS] recommend then that you don't need to take your PrEP. [...] That's when I came off, but they obviously then realized [*partner's name omitted*] probably had HIV for a good 2 to 3 to 4 years and, of course, had quite a high, of what they call, viral load. I believe that's the terminology. I would never have even thought about it and its only because they advised me to. […] They said 'Yeah, yeah, you're not going to be at risk'." (Gay man, White British ethnicity, 25 to 34 years, diagnosed with HIV more than 6 months after the PrEP Impact Trial had ended)

**Communication gaps about sexual health in sexual relationships:** Interpersonal factors could undermine participants' HIV prevention uptake, too. Some participants explained that they did not have in-depth conversations with their sexual partners about sexual health, practices, and prevention, possibly related to a sense of trusting their partner (see more on that below; Motivation: "Identity characteristics determine the risk of HIV"): Although we did not identify that accounts considered under this theme were only deriving from participants with one specific ethnicity, gender or sexual orientation, the accounts of two Black African heterosexual women illustrate that communication gaps often derived from trusting the sexual partner. In one case where the participant had likely acquired HIV from her partner, trust was reflected in the assumption that he was pursuing what she deemed a religious Christian lifestyle (like herself) which for her excluded the possibility that he could have been exposed to HIV:

> "In [African home country], […] they grew up in church things like that and it was the same church that I started with when I came over here, so although he like me at some point in his life he stopped going to church, but I think [inaudible] I didn't realize that he wasn't, […] So, he got to that point in his life when he stopped and then, of course, he was doing things as normal, having girlfriends and stuff. So, just I didn't realise, how, I thought maybe he had stopped, he sort of had a girlfriend and went back to church, but I didn't realize that was not the case, and he was still carrying on living his normal life. So, I didn't realise that's why I never thought of HIV." (Heterosexual woman, Black African ethnicity, 35 to 44 years)

> "[My partner] just said he was fine. Because he said he was fine, I just thought oh [he said he was fine] so we're fine." (Heterosexual woman, Black African ethnicity, 25 to 34 years)

**PrEP is inaccessible:** A few participants reported that they had no way (or difficulty) accessing PrEP. This theme predominantly refers to participants' perception before PrEP was available for free on the NHS via SSHS beyond the PrEP Impact Trial. However, it stresses the impact of restricting access to medication (that may endure) on the considerations of engaging with it (see more on this below; Motivation: "Low self-relevance for PrEP"). For example, participants explained that they considered using PrEP, but that the fact that it wasn't available on the NHS before the end of the PrEP Impact Trial was a barrier:

"[…] at that time PrEP, it's not now but at the time, PrEP was something you had to pay for it in England. And I was, I was actually planning on getting it and because I was actually enquiring about this and planning on getting it, once that became available on the NHS, which I think it became available in September last year. But by then I found out I had had HIV, so there's no point". (Gay man, White British, 25 to 34 years, diagnosed within 6 months after the PrEP Impact Trial).

Some participants tried to get PrEP privately, because it wasn't officially available outside of the PrEP Impact Trial in England at the time. But these attempts were often unsuccessful due to logistical issues (e.g., ordering from abroad, unavailability of PrEP in local area). The restricted access to PrEP until October 2020 and the COVID-19 pandemic occurring at the same time further seemed to make it difficult to get PrEP:

"It was just pandemic, at the very end when it became available. I just, things just for me at that time in my life, everything just wasn't right. I miss the PrEP, you know, things like that […] everything was shot and couldn't go and couldn't go nowhere […]" (Gay man, White British, 25 to 30 years)

**Motivation. Low self-relevance for HIV:** Just over half of participants perceived that HIV was not relevant to them, for example, since they judged themselves as at low risk prior to acquiring HIV, but the reasons for this were manifold. Table 3 provides an overview of the identified subthemes.

One of the two most frequently mentioned reasons were that participants believed that they were not at risk or that they were usually mitigating their HIV risk and were "relatively safe". However, since this wasn't consistently practised, it left participants at risk at times:

"[…] relatively safe, as in, I use protection when I'm having sex and stuff and there's, there's obviously like lapses in that when you, when you go out, have a night out and you drink [alcohol] and stuff, so yeah, but we usually, in terms of sexual health setting, normally relatively safe. (Gay man, White British ethnicity, 25 to 34 years)

The other most mentioned reason was that the risk of HIV acquisition was thought to be contingent on specific personal characteristics, for example, social class, profession, or age:

"I didn't feel any of these things were part of my life quite arrogantly […] you know, when you're [between 50 to 64 years] and you think, you know people, you may still have a few sexually transmitted infections as I [said?] to you. I just

**Table 3. Subthemes of Low self-relevance for HIV theme.**

| Theme | Subthemes (number of participants mentioning underlying accounts) |
|---|---|
| Low self-relevance for HIV (Motivation – Barrier) | • Not at risk or risk of HIV is mitigated (8)<br>• Identity characteristics determine the risk of HIV (5)<br>• HIV is not a problem 'here', or anymore (5)<br>• Discarding information about HIV (4) |

for when they report being single man and gay, you know, I [could've?], locked myself away and but which I didn't want to do, I suppose, I was going to get something but never expecting, don't get HIV." (Gay male, White British ethnicity, 50 to 64 years)

Both subthemes relate to participants attributing the risk of HIV to specific categorizations (i.e., sexual practices, personal characteristics), making stigmatising assumptions about who would 'typically' be at risk (and who wouldn't be) which is a fundamental element of HIV stigma.

Some participants (even those from groups traditionally associated with higher HIV rates), perceived the risk of HIV as no longer relevant today (sometimes thinking HIV did not exist at all anymore) or as less relevant than other sexually transmitted diseases (STIs), which may be more transmissible than HIV, thereby undermining the risk of it:

"Honestly, HIV never crossed my mind. It was always other sexually transmitted diseases like chlamydia or gonorrhoea or all these sorts of things, it's always, like occurs 'Oh I think I should go get tested because I might have chlamydia or something', HIV was never the thing that came to mind, I never I'd never worry about it, just because there's, I didn't really think that was a problem in the UK. You don't really hear like that so many people are diagnosed with HIV, it's not talked about as much." (Gay man, White British ethnicity, 25 to 34 years)

Importantly, others explained that because they didn't think that HIV was relevant to them, they disregarded information altogether:

"No, I haven't [looked for information about HIV], because at the time I wasn't looking, or researching, and if anything came up on my phone or internet, you know, I would never read. I would just, you know, I would discard it straightaway. So no, because I always thought that [in?] my head is never going to happen to me. So therefore, I don't acknowledge it or […], I don't look at them, I just carry on doing my own business because you don't think it's going to happen to you, but now when I see them, I'm more aware of it." (Gay male, White British ethnicity, 50 to 64 years)

This is concerning considering that participants described having gaps in HIV knowledge (see above; Capability: "Gaps in HIV knowledge"); HIV information that is available was apparently only engaged with when it was attributed as (self-)relevant. Self-relevance (Motivation) and knowledge (Capability) are therefore strongly intertwined.

**Low self-relevance for PrEP:** We also identified accounts that evidenced low self-relevance for PrEP specifically. This was again evident in different manifestations. Table 4 provides an overview of the identified subthemes.

The most frequently mentioned reason was low HIV risk perception, which was directly connected to not engaging with PrEP. This therefore connects low self-relevance for HIV with low self-relevance for PrEP:

"Yeah, and also because you think the risks, you know, you know, naively think, you're taking a low risk so you're not so worried, you're not thinking, you know that I've got to check this [PrEP or PEP] out." (Gay man, White British ethnicity, 35 to 49 years, diagnosed with HIV during the time of the PrEP Impact Trial, but before 2020)

**Table 4. Subthemes of Low self-relevance for PrEP theme.**

| Theme | Subthemes (number of participants mentioning underlying accounts) |
|---|---|
| Low self-relevance for PrEP (Motivation – Barrier) | • Low HIV risk perception (5)<br>• PrEP is not a priority (2)<br>• Assumptions about who PrEP is for (2) |

In some cases, we identified that the financial costs for PrEP (i.e., when trying to get PrEP outside of the PrEP Impact Trial) had undermined the judgement of self-relevance for PrEP. This builds directly on the issue of restrictions to medications, identified as a structural barrier (see above; Opportunity: "PrEP is inaccessible"). However, here the focus is the motivational aspect of experiencing exclusion from access to PrEP.

Two participants expressed that taking PrEP was not a priority for them, for example, because it could cause suspicion that they were involved in extradyadic relationships:

"You wouldn't want to find out that, that you wouldn't want them to find out the same thing [taking PrEP] […] it's like, you know, you got to explain that [to your partner] and, you know, then that causes other problems. […] Well, it is. It's bad for, you know, when you're married." (Gay man, White British ethnicity, 35 to 49 years, diagnosed with HIV during the time of the PrEP Impact Trial, but before 2020)

Finally, two participants made specific stigmatising assumptions about who PrEP was for – and consequently who it wasn't for, thereby mirroring the stigmatising assumptions made about HIV mentioned above:

"PrEP is [inaudible], but I think that the people that want that go on that is because they're playing around." (Gay man, White British ethnicity, 50 to 64 years, diagnosed with HIV during the time of the PrEP Impact Trial in 2020)

**PrEP scepticism:** Some participants explained that they were sceptical about using PrEP because they were concerned about the side effects or perceived it as difficult to adhere to a daily oral tablet. The fear of side effects motivated some participants to keep using condoms. One participant explained that they at first understood side effects as a 'price to pay' in order to prevent HIV:

[…] the basic information available for everyone, […] will say they more about what is going to happen when you take it here. I mean, everyone I believe knows that 'OK, to prevent you to get infected with HIV, but you pay the price for it.' Yeah, because that's how I read this information at the first place. (Gay man, White other ethnicity, 30 to 35 years)

Despite PrEP side effects commonly being mild and temporary, perceiving these as otherwise, for some was a major barrier to the uptake of the prevention and is, therefore, likely connected to being misinformed or uninformed about it (which many participants described). Consequently, the fear of side effects, connects people' motivations to their capabilities (i.e., lacking knowledge), and missed opportunities (not being informed), as already indicated in the previous sections.

Some participants described it as potentially difficult to adhere to a daily oral tablet which also emphasises a potential lack of knowledge about event-based PrEP:

"[PEP] is a better one, because if you have sex, you can take the pill, like the 'morning after pill' if you want to, and you're not constantly taking a pill every day for it to kick in. I can tell you it depends if you've had sex and you don't get it, you know, and I think that's probably better". (Gay man, White British ethnicity, 50 to 64 years, diagnosed with HIV during the time of the PrEP Impact Trial but before 2020)

**PrEP stigma:** Some participants reported that they perceived getting PrEP to be associated with stigma. In one case, this referred to being 'outed' as a sex worker, but predominantly, it referred to the presumed perceptions of healthcare professionals. One participant explained that because they identified as bisexual at the time, they assumed that healthcare staff would not categorize them as in need of PrEP (because they would possibly be read as heterosexual and therefore at lower risk), indicating that the identification of PrEP was perceived as being exclusive to gay men (cf. Opportunity: "Unmet needs in healthcare settings to learn more about and to receive PrEP"):

"I think the clinicians, they always do thinking about guys or people which they are in more risk than I was at that time. So, I felt like 'OK, I'm not going to bother because probably I won't get it anyway'". (Gay man, White Other ethnicity, 35 to 44 years, diagnosed with HIV during the time of the PrEP Impact Trial in 2020)

The example highlights that perceptions about what other people may think can undermine the self-relevance of PrEP (see above; "Low self-relevance for PrEP"). In another example, a non-UK born participant recalled their experiences with stigma when trying to access it in their home country, thereby also reflecting that the situation there was different compared to England:

"It's different here than it was in [African country], because of the perception, you know, people had and stuff, and so once you go in there [clinic], most people, you're scared you might see someone you know. So, you might be coming up on that clinic or what are you doing with that, so I do feel, you know, are you ill, or are you already positive, so to avoid not to give people something to talk about. So, it was easier not to go." (Gay man, Black African ethnicity, 25 to 34 years, diagnosed with HIV during the time of the PrEP Impact Trial in 2020)

**Existing HIV campaigns are too selective:** Whilst some participants had pointed out that there was a lack of current HIV campaigns (see above; Opportunity: "Unmet needs to learn more about HIV"), the accounts of two participants illustrated that some HIV campaigns were perceived as too selective. Participants perceived these as either leaving some people out of the campaign – thereby undermining identification with (and intention to act upon) the content of the campaign;
"Because the campaign, when you [look at] it, it would just be like, I think it was like, it was more older gay men. When I looked at the campaign. I feel like [I?] was really [cross at it?]. I wouldn't really be like 'Oh that screams me. That screams that I need to go and get tested.' And you just hear someone holding the finger, and you wouldn't really understand the thing, I think." (Gay man, White British ethnicity, 25 to 30 years)
or conversely being too targeted – thereby leaving the impression of stigmatising a community (i.e., being the only group associated with HIV):

"I'll tell you one thing that used to put me off. When I used to see some adverts actually, so being African, so before I'm just, you know, like sometimes when I see some adverts, oh, I think I use to see some adverts and they would say 'Put your finger up to HIV' and things like that, I'm just telling you what I would see in my head, I would always see a Black person, an African Black person right, so you know it was more targeted on them. […] So, it puts you off and, and you sort of see the person as they're condemned, or things like that" (Heterosexual woman, Black African ethnicity, 35 to 49 years)

Although both accounts referred to HIV testing campaigns, these were made in response to the interviewer question about noticing wider HIV campaigns (cf. Opportunity: "Unmet needs to learn more about HIV"). They therefore added to the question of how HIV campaigns can reach those most at need of HIV prevention methods other than testing, including PrEP.

**Facilitators**

**Capability. Gaining knowledge about PrEP:** Just under half of participants explained that they had learned about PrEP through a variety of sources which included other people (e.g., peers and healthcare professionals), as well as specific media outlets (e.g., social media, TV, or leaflets):

"And so, I did know about PrEP and PEP. [...] Yeah. So, I did know about those. And you talk about, I talked about it with my friend." (Gay man, White British ethnicity, 25 to 34 years, diagnosed with HIV within 6 months of the end of the PrEP Impact Trial).

Three participants whose accounts were coded under this theme were also coded as presenting gaps in PrEP knowledge (see above; Capability: "Gaps in PrEP knowledge"). The reason for this overlap was participants explaining that they had 'some' knowledge but not enough to enable them to take PrEP. Among participants whose accounts were coded under this theme, only one participant had actually used PrEP, demonstrating that knowledge does not automatically translate into action.

**Motivation. Wanting to stay safe and take control of one's health:** Whilst sexual relationships seemed to sometimes hinder engagement with HIV prevention methods (see above; Opportunity: "Communications gaps about sexual health in sexual relationships"), some participants explained that previous undesirable sexual encounters made them consider taking PrEP. Others explained that they felt particularly fearful and cautious regarding sexual activities and the risk of HIV, for example, when entering their first same-sex relationship as a gay man. The consideration to take PrEP was therefore enhanced through a change in participants' risk perception as a function of how relevant PrEP was, and consequently wishing to take control of their health. One participant illustrated that by saying that once they had realised that a past sexual partner had not shared their HIV status, they got more cautious and fearful and wanted to start taking PrEP:

> "And I thought October [2017 – the beginning of the PrEP Impact Trial] is not far away, it's like what nine months away so I thought I ask 'Could I be put on some kind of list to you know to just wait or register for it to get it once its available' because it's so close to get it free, why would I pay to get it. So it was like a whole process, like a complex process. So, I thought, I am happy, I am [HIV] negative, I am not doing anything really wrong, and I don't have any other STI so I am happy as well. So that's how it went. And then I came back in March, maybe actually because I came back, I wanted to, again, about the PrEP, but then I thought, 'OK, I'm going to request and I'm going to pay for it.' [*Interviewer asks whether that was because the participant saw more men*] Yes. Exactly, only because only because of that. On top of that, I find out that one of the men I've been seeing at that time, he lied to me. So, he'd been [HIV] positive, but he never mentioned that. But he was undetectable. Yeah, but it's nothing like that is it doesn't make any difference. It was just like; I saw people like lying [about their HIV status]? So, I thought, 'OK, he was undetectable. Another one might not.' And so on. So, it's better to do something about it". (Gay man, White Other ethnicity, 25 to 34 years, diagnosed with HIV during the PrEP Impact Trial in 2020)

## Discussion

Medical advances in HIV treatment mean that those on treatment can achieve HIV viral suppression – an undetectable viral load – and, therefore, cannot pass the virus on and live a healthy life. However, not everyone is diagnosed nor virally supressed, therefore, HIV transmission continues. Thus, prevention remains a key part in working towards ending transmission. Exploring the barriers to HIV prevention and PrEP use among people who now live with HIV is inevitably retrospective. However, understanding these challenges can reveal strong narratives of missed opportunities for HIV prevention. This can help influence people not living with HIV to make informed health decisions. In this study, we applied a systematic framework – the COM-B model [18] – to explore the previous barriers and facilitators to consideration of using PrEP, use, as well as overarching factors constituting barriers and facilitators to HIV prevention more broadly data among people now living with HIV. Our results revealed that participants prior to their HIV diagnosis had (and indeed still have) knowledge gaps about HIV and PrEP specifically. Mental health issues were also a barrier to their engagement with HIV prevention strategies which aligns with previous research conducted among people not living with HIV [10,15,19,35]. Importantly, we found that poor mental health not only impacted momentary decision-making processes (i.e., risk taking) but also a broader sense of self that could make participants feel ambivalent about themselves and their health which consequently also impacted engagement with, for example, clinical outreach, making this a holistic problem.

Recent national surveillance data showed that in 2023, in English SHSS, 85.2% of GBMSM were identified as in need of PrEP whilst this was only the case for 62.2% of heterosexual and bisexual women and 60.8% of heterosexual men [9]. Previous research found that found that eligibility criteria for PrEP constituted one of the strongest provider barriers for Black women in the UK to engage with PrEP [17]. Although structurally not being recognised to be in need of PrEP [36] minimised chances that people would learn more about HIV or PrEP, in our sample we found that this was predominantly identified to be an issue by gay men rather than Black women. However, this could reflect sample bias (see below; Limitations). We did not identify themes that were unique to Black ethnicity participants. However, our study confirmed the presence of previously found barriers to HIV prevention or PrEP among Black people [13] such as religious factors and stigma.

Whilst previous research found that interpersonal barriers existed in the form of concerns about sexual infidelity or a lack of partner support [10], we identified that a lack of communication between sexual partners undermined the uptake of PrEP, often connected with a sense of trusting partners were sexually at no or low risk or making inaccurate assumptions. These two findings do not mutually exclude each other but rather go hand in hand; where there is a lack of partner communication, there is likely concerns about how engagement with PrEP could be perceived by partners. Previous research had also identified that access to PrEP and financial costs were strong structural barriers to the uptake of PrEP [10]. Whilst this is in part due to different healthcare systems (most reviewed studies were conducted in the U.S. where PrEP is not free), it is comparable to the UK context at the time when the interviews of our study were conducted; as previously mentioned, the interviews were conducted in part during the PrEP Impact Trial and none of our participants had taken part in the trial. Consequently, some participants felt that PrEP was inaccessible to them (the implications of costs are discussed in more depth below). Finally, learning about the benefits and efficacy of PrEP had previously been identified as facilitators to the uptake of PrEP [10,17]. Our study is in line with that by not only showing that learning about PrEP increased the consideration of taking PrEP, but we also identified a strong unmet need to learn more about HIV in general and about PrEP specifically.

However, the most impactful variable which we identified as critical for the engagement (or disengagement) with HIV prevention strategies was (not) perceiving HIV prevention strategies to be relevant to oneself. Previous research had highlighted how a lack of PrEP knowledge among Black women in the UK can undermine HIV risk perception and PrEP relevance [17]. In our study, we found that even where HIV or PrEP information had been accessible to participants, this could still be the case. According to Parsons et al. [16], people are 'pre-contemplative' of PrEP, (i.e., in the first phase of the motivational PrEP cascade), if they do not view themselves as PrEP candidates. Coukan et al. [15] did not categorize the barriers and facilitators that they had identified in their research into this phase because non-engagement with PrEP was understood as the status quo ("the default", p. 907). We believe that being pre-contemplative indeed constitutes a major barrier to the use of PrEP. Behavioural models concerned with coping strategies (i.e., whether and how people deal with a threat), commonly treat self-relevance as the precursor of any action (or inaction, respectively) [37,38]. For example, the premise of Protection Motivation Theory [37] is that people are motivated to protect themselves from a threat as a function of how they appraise the threat and their coping options. Threat appraisals concern beliefs about risks which are informed by the perception of severity, susceptibility, or vulnerability. Only where the threat is appraised as high enough, will people engage in coping appraisals, concerned with the actions to prevent such threats. In other words, people must assess a threat as sufficiently and personally meaningful and oneself to be at risk (i.e., 'the threat of HIV is severe, and it is severe to me', 'HIV can harm, and it can harm me', 'I am vulnerable to HIV'), to consider acting upon such information. Consequently, if people are pre-contemplative of PrEP, this constitutes a barrier to the uptake of PrEP because PrEP is not considered sufficiently self-relevant (yet). In our study, most accounts among all the motivational barriers we identified related to a perceived low self-relevance for PrEP and HIV risk in general. Whilst we need to consider the U.S. American context in which the motivational PrEP cascade model was first developed (PrEP is available in the U.S. only with medical insurance [39]), the (in-) accessibility of PrEP, particularly prior to its availability in SSHS as a free NHS intervention, allows us to compare our findings to the model's assumptions. Financial costs negatively impacted self-relevance for

PrEP in our sample, too, but it was not only influenced by financial considerations. Low self-relevance was especially evident when participants did perceive themselves to be at low risk (compared to others), perceived that PrEP or HIV relevance did not align with their identity (but rather that of others), and did not believe that HIV was a problem "here" (but elsewhere). We identified explicit mentions of HIV and PrEP stigma that constituted barriers to the uptake of PrEP. However, the comparisons we identified in participants' accounts about who is perceived to be at risk of HIV and therefore in need of PrEP and who is not, also reveal stigmatising assumptions. Concurrently, this reveals coping mechanisms that seek to avoid stigma [40], however, they also perpetuate stigma. In other words, HIV and PrEP stigma can foster both an inaccurate risk perception and poor self-relevance, possibly even despite accessible HIV and PrEP information; individuals seek to distance themselves from the stigma of HIV and the associated need for prevention (including PrEP) thereby reinforcing negative perceptions towards people they presumed to be associated with HIV. This not only decreases an individual's ability to prevent HIV but further reinforces HIV stigma at a societal level. We believe that our findings of substantially more barriers to PrEP than facilitators to consideration of PrEP use plus the majority of participants not having used PrEP, is indicative of HIV stigma.

Overall, this study concludes that structural opportunities and self-relevance, can influence people' capabilities to know about PrEP and motivations to take action and prevent HIV, both of which we identified as facilitators to considering PrEP. This illustrates how the behaviour change domains are interconnected and how all of these must be addressed in order for behaviour change to occur.

### Practical implications

Many non-medical health interventions approach behaviour change with an enhanced effort to promote knowledge of healthy behaviours and/or reduce the negative consequences of unhealthy behaviours. However, knowledge about HIV prevention and PrEP alone is often insufficient for behaviour change [41]. Our results underline this, and we suggest that in addition to enhancing the knowledge of PrEP, enhancing accurate risk perceptions and assessments of self-relevance are needed. Wider societal discourse as well as environmental provisions are necessary to increase the use of PrEP. On an individual level, we believe that skills training or tools to identify HIV risks and negotiating safer sexual health needs and agency in sexual relationship is a meaningful complementation to HIV and PrEP knowledge. We recommend a greater integration and provision of training materials on relationships, sexual health and HIV prevention knowledge and negotiation skills in schools, universities, and local communities, for example using resources provided by community organisations and charities specialising in sexual health and HIV (e.g., SRE forum, Brook, Lovetank, or HIV prevention England) [17,42–44]. On a structural level, considering the role that mental health played in the engagement with HIV prevention and PrEP, we suggest that SHSs services should endeavour for frontline staff to have Mental Health first-aid training and to ensure each clinic has a PrEP prescriber who is trained in general mental health and wellbeing [e.g., becoming accredited with the HIV Confident charter 45]. Integrating mental health screening, treatment and substance use support into PrEP delivery programs, and even expanding delivery to community pharmacies, mental health settings and primary care could also address the role of mental health and PrEP delivery barriers [46,47].

In 2019, the Department for Health and Social Care HIV Prevention Innovation Fund funded 14 HIV community-led interventions, targeting higher risk and underserved communities (e.g., women, Black people, young men, people who were incarcerated) [48]. Six initiatives explicitly addressed PrEP. The evaluation of these projects [49] found evidence that HIV and PrEP awareness and knowledge as well as usage increased. We therefore suggest continuously evaluating and reviewing HIV prevention interventions and campaigns for their effectiveness and ability to also appeal to communities that historically and currently do not identify themselves as at risk of HIV, as well as those necessarily usually targeted most at risk. However, our results also indicated that HIV prevention campaigns may need to be more visible and appeal to a broad range of groups to navigate visibility and therefore relevance to 'at higher risk' groups and concurrently avoiding stigmatising these groups by featuring them only. In 2022, NHS England began widespread implementation of bloodborne

virus (BBVs: HIV, Hepatitis B and C) opt-out testing in emergency departments (ED) in areas of high HIV prevalence This means that patients in EDs are automatically tested for BBVs unless they opt out [50]. It would be useful for SSHS to introduce opt-out testing for HIV and discussion of PrEP. The recent evaluation of the ED BBV opt-out testing suggested that opt-out methodology has been successful in testing people who wouldn't otherwise be tested in other settings [51]. This methodology has therefore the potential to reduces bias and stigma, to normalise HIV testing, and to increase HIV self-relevance and accurate risk-based assessment across more communities.

Our results suggest that people would benefit from a wider provision of PrEP access, and PrEP and HIV education beyond at SSHS alone. New guidelines on PrEP availability were published recently [52] which recommend widening access to PrEP beyond the inclusion criteria used for clinical trials, in order to achieve equity and reduce inequalities. PrEP is now recommended, among others, to all those who would benefit from reduction in HIV risk including people requesting PrEP, regardless of gender and sexual orientation. Widening the guidelines and the provision to primary care and pharmacies may also contribute to the normalisation and awareness of PrEP [36,45]. The results also suggest the usefulness of injection-based long-acting PrEP as an alternative to daily or event based oral tablet, which is has now been approved in England but is yet to be made available [8].

## Limitations

The main limitation of this study is its sample bias. Most participants identified as White, British, male and gay. Heterosexual and Black people, as well as women (and the intersection thereof) are under researched in the context of HIV prevention, and PrEP specifically [15]. Despite efforts to over-sample people with these characteristics in this study, our sample underrepresented these groups so that it still consists of predominantly White gay men due to higher uptake in this group. For future research to more effectively recruit from under researched communities, trusted relationships with various communities, co-production, and broader societal and structural change are required. Even through the compensation at the time (£25) is in line with the National Institute for Health and Care Research (NIHR) guidelines for payment rates [53], we believe that a higher payment rate would facilitate the recruitment success considering the variety of socio-economic backgrounds of participants.

Sharing common personal characteristics (e.g., gender, sexuality, or ethnicity) with interviewees can enhance trust and rapport, making them feel more comfortable and understood [54–56]. However, cultural competence and humility are as important in navigating interviews, and as mentioned, in some cases, participants welcomed the fact that the interviewers were of a different ethnicity and opened up because of perceived HIV stigma in their community of origin. Going forwards, we recommend providing a more diverse interviewer panel regarding gender, ethnicity, language and socio-economic class and for participants to have the choice about by whom they prefer to be interviewed. We also suggest that interview locations and timings may need to be flexible as there were concerns about participants being unable to receive phone calls (e.g., due to financial circumstances, work shift patterns etc).

We further acknowledge that the timing of the interviews may have impacted the study's insights. We interviewed participants post-diagnosis, and their perceptions and experiences therefore lie in the past and may be reconstructed in part. Relatedly, some time has passed since the interviews were conducted (2021, 2022). Considering that the experiences with PrEP we made pre-diagnosis (thus, in most cases, before 2021 and 2022), some accounts refer to situations that may no longer be relevant. This was particularly evident in discussion of PrEP access in England. Many participants explained that they perceived PrEP as inaccessible. Participants for this study were recruited and interviewed during and after the PrEP Impact Trial (22). The perceived barriers (and facilitators) to PrEP were therefore influenced by perceived and real (in-) accessibility of PrEP prior to its availability in sexual health clinics in England. Considering that recently the guidelines for PrEP have changed and now recommend a wider eligibility [52], these insights may therefore be limited in their applicability today. That said, many participants suggested that PrEP needs to become even more accessible in future [e.g., in pharmacies; 36]. Finally, our qualitative research design cannot establish causality, and we recommend

following up on our findings with quantitative designs that can explore the causal effect of structural opportunities and self-relevance on the uptake of PrEP.

## Conclusion

Barriers and facilitators to enact HIV prevention, including PrEP, are well researched among people not living with HIV [10,12,15,19,35], but to the best of our knowledge, research on (dis-) engagement with PrEP that considers the reflections of people who are now living with HIV is missing [20]. In this study, we applied the COM-B model onto data from semi-structured interviews with people living with HIV, conducted in the context of enhanced surveillance of people with recently acquired HIV. The analysis explored the barriers and facilitators to considering PrEP use, including overarching factors of barriers and facilitators to HIV prevention more broadly. We believe that the reflections of people who are now living with HIV could identify barriers that were particularly influential in decision making and access to prevention strategies, including PrEP. This may produce strong narratives by highlighting the relevance of PrEP with a 'Things I wish I had known/ done/ or been provided with' approach to HIV prevention options.

Whilst we found that our results generally aligned with research previously conducted among people not living with HIV, we established that missed structural opportunities, HIV stigma and low self-relevance for HIV and PrEP undermined the considerations to use PrEP. We conclude that HIV (and by extension PrEP) understanding, accurate risk perception and self-relevance are fundamental to the consideration of using preventive medication, but that societal and structural factors that consciously and unconsciously reinforce HIV stigma undermine these. This supports the Combination Prevention Approach to HIV but also highlights that more tailored work is needed. We suggest that further research on PrEP issues is needed and provide some research and practical recommendations.

## Supporting information

**S1 Text. Topic guide.**
(DOCX)

**S2 Text. Codebook.**
(DOCX)

## Acknowledgments

We want to thank the following people for their contributions to this study: All participants and contributing clinical sites who generously gave their trust, time and insight, Adamma Aghaizu (initial conception/ author), Nicky Connor (initial conception), Valerie Delpech (initial conception), Amber Newbigging-Lister (initial conception), Helen Corkin (interviewer/ author), Vicky Gilbart (interviewer), Alison Brown (revision), Catherine Lowndes (revision), Veronique Martin (revision), Freya Mills (revision), Riinu Pae (revision), Avelie Stuart (revision), as well as members of the steering group.

## Author contributions

**Conceptualization:** Adamma Aghaizu.

**Data curation:** Dolores Mullen, Ammi Shah.

**Formal analysis:** Carina Hörst, Hannah Kitt, Dolores Mullen.

**Investigation:** Helen Corkin.

**Methodology:** Carina Hörst.

**Project administration:** Carina Hörst.

**Supervision:** Adamma Aghaizu, Clare Humphreys, Tamara Djuretic.

**Writing – original draft:** Carina Hörst.

**Writing – review & editing:** Carina Hörst, Hannah Kitt, Helen Corkin, Dolores Mullen, Ammi Shah, Adamma Aghaizu, Clare Humphreys, Tamara Djuretic.

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
