## [Decision Letter · Decision Letter 0]

15 Oct 2025

PONE-D-25-26292Retrospective challenges to pre-exposure prophylaxis (PrEP) use among people living with HIV – A qualitative analysis using the COM-B framework.PLOS ONE

Dear Dr. Hoerst,

Thank you for submitting your manuscript to PLOS ONE. After careful consideration, we feel that it has merit but does not fully meet PLOS ONE’s publication criteria as it currently stands. Therefore, we invite you to submit a revised version of the manuscript that addresses the points raised during the review process.

**Please address all of the comments from the reviewers prior to resubmission; thank you.** ==============================

We look forward to receiving your revised manuscript.

Kind regards,

Douglas S. Krakower, MD

Academic Editor

PLOS ONE

Journal Requirements:

2. We noted in your submission details that a portion of your manuscript may have been presented or published elsewhere. “A publicly available UKHSA report has been published which refers to the SHARE initiative and the topic examined in our manuscript. However, this presents a high-level overview alongside other surveillance initiatives and topics. It is not academic in nature, nor does it discuss the research question or methodology in-depth. We confirm that the manuscript submitted is original work and has not been published, nor is it under consideration, elsewhere.” Please clarify whether this [conference proceeding or publication] was peer-reviewed and formally published. If this work was previously peer-reviewed and published, in the cover letter please provide the reason that this work does not constitute dual publication and should be included in the current manuscript.

Reviewers' comments:

Reviewer's Responses to Questions

**Comments to the Author**

1. Is the manuscript technically sound, and do the data support the conclusions?

Reviewer #1: Yes

Reviewer #2: Yes

2. Has the statistical analysis been performed appropriately and rigorously? 

Reviewer #1: Yes

Reviewer #2: N/A

3. Have the authors made all data underlying the findings in their manuscript fully available?

Reviewer #1: Yes

Reviewer #2: Yes

4. Is the manuscript presented in an intelligible fashion and written in standard English?

Reviewer #1: Yes

Reviewer #2: Yes

5. Review Comments to the Author

Reviewer #1: This is a well written qualitative study examining challenges to being on PrEP for persons who had recently been diagnosed with HIV. There have been many qualitative studies conducted on PrEP barriers and facilitators among persons without HIV who are at risk. The authors make a good case that this study fills a gap as it evaluates individuals who have recently acquired HIV to see if retrospectively they can recognize barriers/missed opportunities for PrEP. In this respect, there is some novelty/added value to this paper that is on a topic that is relatively highly researched. The paper also uses the COM-B framework which is a strength. A major limitation is that most participants were white men so there was limited information gathered from the populations that appear to be "falling through the cracks" (as they outline in the introduction), namely nonwhite persons and women. Some minor suggestions to strengthen the manuscript:

1) While poor mental health is identified as a barrier, there is a lack of acknowledgement/discussion about the role of substance use despite this being mentioned by some participants in quotes.

2) The low participation rate is something that should be acknowledged and discussed. Was stigma/shame associated with new diagnosis a barrier to participation? I could also imagine that discussions about missed opportunities for PrEP might bring up feelings or remorse or guilt. Was there any attempt to mitigate or address this in the research procedures?

3) There was relatively little in the discussion (until the end) about the COM-B framework. Perhaps there could be earlier discussion about the strengths/utility of that framework?

4) Throughout there are some abbreviations that are confusing that may relate to citation software. For example, see page 7 "cf. 19"

5) Page 9, line 200 (under "Participants" should be edited to say either "history of injection drug use" or "history of injecting drugs" and later in the sentence there is an extra "a" that should be removed ("and one a had a history of imprisonment").

Reviewer #2: Thank you for the opportunity to review this interesting study. I believe that this study is important and provides helpful insight regarding PrEP delivery in the UK. This is a great manuscript but there are areas that can be strengthened.

Overall, throughout the manuscript I would replace the “HIV-negative people” with “people not living with HIV.” I believe this would be less stigmatizing because the reverse of HIV negative is HIV positive. I would also drop the word infection after HIV throughout. It’s not necessary.

I would suggest adding the Table numbers and titles directly in the tables.

Abstract

I believe that the description of the participants could be clearer and to the point. It mentions including “gay men, heterosexual people, Black and female people.” I would suggest a change to something like “including a diverse group of individuals.”

Introduction

Can you please provide more information regarding the types of PrEP are available in the UK? The tenofovir/emtricitabine implies Truvada and Descovy. There is a line that mentions use during event-based sex but Descovy has not been approved for that. Can you please clarify?

Can you also write out specialized sexual health services the first time it’s mentioned in the intro? Can you also give a few words/line to describe it for the international audience not as familiar with the UK systems?

Can you also discuss briefly the eligibility criteria for PrEP in the UK? This will aid with understanding the discussion more.

Can you please provide data regarding the highest risk age ranges for new HIV diagnoses in the UK? I think this would tie well into the results.

The literature specifically regarding those of Black ethnicity and barriers to PrEP use could be stronger. Would suggest additional references.

Methodology

I believe the methodology is appropriate.

Could you please elaborate in the manuscript about the kind of handover between the clinicians who consented the participants to the researchers? Did the participants know anything about the interviewers prior? If so, what?

It would be nice to see all the demographics in a single Table 1. For example, the employment part is not in the table.

Is there a reason level of education was not collected/reported?

Some details regarding the qualitative methods are missing. Could you please address these in the manuscript?

In the data collection part, could you please elaborate more on the non-participation? The goal was 50 participants. There were 45 consented but only 26 interviewed. Did people not pick up the phone? Were researchers not able to find the additional 19 participants?

Were the interview questions piloted?

Were field notes taken during the interviews?

Were the participants alone during the interviews or were potential non-participants around?

Were the participants able to review the transcripts and comment and/or correct the statements?

In the data analysis, could you please provide the NVIVO version, a brief description of the software and a citation for those not as familiar with qualitative analysis?

With 26 participants there was likely data saturation. Could you please comment on data saturation?

Could you please be more specific about who the 2 interviewers were? I would also add that the interviewers were female in this section rather than later.

Results

Capability, Gaps in HIV Knowledge, and Gaps in PrEP Knowledge are not in bold but the rest are.

In the intro, the high numbers of new cases of HIV in those with Black ethnicity was emphasized, were any of the barriers or facilitators more prominent among those participants?

Discussion

The discussion could be strengthened with more discussion regarding the opportunity barriers found and existing literature.

In the paragraph from lines 626-633, I think this would be a good opportunity to mention that the eligibility criteria has since been updated. There have been new guidelines published in 2025.

Any suggestions regarding provider level interventions to improve the unmet healthcare needs and communication gaps based on this data?

Given the role of mental health in PrEP delivery, do you have any additional suggestions to address this barrier besides solely the holistic person-centered approach?

In lines, 714-717 I would remove “provided every other” month unless you specifically would like to refer to Cabotegravir. Lenacapavir is now approved in the US and the dosing is twice yearly.

Limitations

Can you please elaborate on why the sample bias is symptomatic of HIV stigma?

I would add to the limitations that the PrEP guidelines for eligibility have changed which so perceptions regarding access and self-relevance may be different.

Conclusion

Would remove extra letter from line 764.

Reasonable conclusion.

6. PLOS authors have the option to publish the peer review history of their article (what does this mean? ). If published, this will include your full peer review and any attached files.

**Do you want your identity to be public for this peer review?** For information about this choice, including consent withdrawal, please see our Privacy Policy .

Reviewer #1: **Yes:** Judith Tsui

Reviewer #2: No

---

## [Author Response · Author response to Decision Letter 1]

9 Dec 2025

Letter to the editor: Revision of “Retrospective challenges to pre-exposure prophylaxis (PrEP) use among people living with HIV – A qualitative analysis using the COM-B framework.” (PONE-D-25-26292)

Dear Prof Douglas S. Krakower,

We thank you for inviting us to submit a revised version of our manuscript titled as above.

We structured this letter in a way that we first address editorial queries, followed by the reviewers’ suggestions split by the degree of revision. We present and discuss this in the order of the manuscript sections for your convenience and provide the relevant page numbers and revised excerpts (page numbers refer to the manuscript with no shown tracked changes).

Please note that we also made minor improvements ourselves as part of the revision process.

We believe that the reviewers’ suggestions and our revisions thereof have substantially strengthened our manuscript, and we hope that it is to your satisfaction.

Kind regards,

Carina Hoerst

---

Editorial queries

The manuscript and title page have now been revised to be in line with the PLOS ONE style requirements. Please note that level 4 headings are not identified in the requirements. We therefore left them as they were.

---

2. We noted in your submission details that a portion of your manuscript may have been presented or published elsewhere. “A publicly available UKHSA report has been published which refers to the SHARE initiative and the topic examined in our manuscript. However, this presents a high-level overview alongside other surveillance initiatives and topics. It is not academic in nature, nor does it discuss the research question or methodology in-depth. We confirm that the manuscript submitted is original work and has not been published, nor is it under consideration, elsewhere.” Please clarify whether this [conference proceeding or publication] was peer-reviewed and formally published. If this work was previously peer-reviewed and published, in the cover letter please provide the reason that this work does not constitute dual publication and should be included in the current manuscript.

The report (titled ‘HIV prevention barriers and facilitators: findings from qualitative interviews among people diagnosed with HIV, March 2021 to July 2022’ which can be accessed here: https://www.gov.uk/government/publications/hiv-prevention-barriers-and-facilitators-qualitative-findings/hiv-prevention-barriers-and-facilitators-findings-from-qualitative-interviews-among-people-diagnosed-with-hiv-march-2021-to-july-2022) has not been academically peer-reviewed and not undergone a formal academic publishing procedure. It was published on GOV.UK (an official UK public sector website) where high-level UKHSA research reports sometimes get published before more detailed and formal submission to an academic journal. We do not consider this report to be a dual publication as the subject of the GOV.UK report is much broader than our manuscript submitted to you; SHARE and PrEP mark only some out of many aspects discussed in this piece. SHARE originally explored a range of different HIV preventions (e.g., condom use, HIV testing) and experiences of people affected. The GOV.uk report constitutes a high-level overview of these.

---

The data that support the findings of this study have been assessed by the UK Health Security Agency Office for Data Acquisition and Release as having sensitive personal information and are therefore not publicly available to protect participant privacy. However, some summaries of the data may be available upon reasonable request from the UKHSA. Requests can be directed to DataAccess@ukhsa.gov.uk.

Higher degree revision

Intro

R2: The literature specifically regarding those of Black ethnicity and barriers to PrEP use could be stronger. Would suggest additional references.

We think that this is a sensible suggestion by the reviewer, and we integrated further research on barriers to PrEP among Black people. However, since the majority of our participants were in fact not of Black, but of white ethnicity GBMSM, we balanced this with the already presented barriers and facilitators to a variety of populations, including GBMSM. Specifically, we now mention the included populations in Antonini et al., 2023 systematic review (which included GBMSM and Black people), as well as that it included African-based studies (despite featuring predominantly U.S. studies). We have singled out a couple of insights from these and additional studies that illustrate how cultural influences can negatively affect the uptake or continuation of PrEP. We removed the references Chebet et al., 2023 and Nabunya et al., 2023 for a lack of additional high-level insights that weren’t already discussed and for risking to shift the focus too far away from our sample composition (see above). Subsequently, we discussed how cultural factors can impact Black people who migrate to other countries (including the UK) and that therefore culturally sensitive communication is critical. Finally, we discuss a UK systematic review which included a variety of populations (including GBMSM and Black people) and highlight that a policy focus on GBMSM can be an objective and subjective barrier to the uptake of PrEP among Black heterosexual people in the UK, especially Black women. We hope that this is to the satisfaction of the reviewer:

pp 3 - 5

“The barriers (and facilitators) to HIV prevention, and PrEP specifically, among different populations at risk of HIV are well researched. A systematic review of quantitative, qualitative and mixed-methods studies (predominantly conducted in the U.S. and African countries) identified individual, social and interpersonal and structural barriers to the uptake of PrEP among people (including GBMSM, young adults, trans and ciswomen, sex workers, Black and ethnic minoritised people) who were currently or previously taking PrEP [10]. Individual barriers related to, for example, concerns about medical side effects, difficulty adhering to daily tablets, perception of low individual risk for HIV, mental health problems and unplanned sexual encounters (when participants were using “event-based” PrEP). Social and interpersonal barriers included PrEP stigma (e.g., fearing that PrEP uptake would be associated with certain stigmatised sexual behaviours and identities), and a lack of partner support (e.g., fearing to be viewed as sexually not monogamous). Structural barriers included limited access to PrEP, long waiting times, negative or insensitive attitudes from healthcare workers and high financial costs. Insights from the African continent reveal the influence of cultural factors such as family pressures impacting PrEP uptake [11], or negative parental influence, for example, among young people in Uganda, Zimbabwe or South Africa where sex at young age and before marriage is culturally disapproved of [12]). Evidence suggests that cultural and religious factors that can shape perceptions and willingness to use PrEP (e.g., discomfort discussing sex and sexual health, misconceptions that PrEP is a treatment for HIV, or concerns about sexual fidelity) often persist when individuals migrate to other places or countries, as seen among communities in the African diaspora settling the U.S. or the UK [13]. Therefore, culturally-competent healthcare provider–patient communication has been shown critical in empowering women of Black African and Caribbean descent living in the UK to use PrEP [14]. A systematic review from the UK synthesised studies that employed quantitative, qualitative, and mixed-methods approaches, which focused on people not living with HIV – primarily GBMSM, including a small number of Black participants, as well as women (including trans women), Black and ethnic minority groups, and people who inject drugs – who had accessed SSHS [15]. Identified barriers and facilitators to the uptake of PrEP greatly overlapped with the aforementioned. However, the authors used a behaviour change framework – the Motivational PrEP Care Continuum [16] specifying five steps from ‘pre-contemplation’ to’ PrEP maintenance’ stage – which provided a more systematic overview of where the barriers and facilitators are located in individual’s PrEP journey. Whilst lack of PrEP knowledge and self-perception of HIV risk were, for example, categorised in the earlier stages on the continuum, perceptions of PrEP stigma, eligibility criteria and access fell into the later stages. A follow up qualitative study, conducted by the same first author [17] explored the barriers and facilitators to PrEP uptake specifically among Black African, Black British and Black Caribbean women in England. They employed the widely used COM-B model [18] – assigning barriers and facilitators to the model’s domains of individual capability and motivation, as well as social and environmental opportunities. The most significant individual-level barriers to PrEP uptake were gaps in knowledge, largely stemming from limited information provided to Black women. At the provider level, restrictive policies around eligibility, exclusive provision through sexual health services, and priorisation practices were identified as key barriers. Conversely, enhanced PrEP education, broader policy access, community engagement, and trusted messengers promoting bodily autonomy and empowerment were identified as strong facilitators.”

Methods

R1: 3) There was relatively little in the discussion (until the end) about the COM-B framework. Perhaps there could be earlier discussion about the strengths/utility of that framework?

We have now signposted the utility of the COM-B model earlier in the introduction, and further outlined its strengths in the Methodology section (Design and theoretical framework). We begin the Discussion now by highlighting our systematic approach to exploring the barriers and facilitators to PrEP by employing the COM-B model.

p. 5

“They employed the widely used COM-B model [18] – assigning barriers and facilitators to the model’s domains of individual capability and motivation, as well as social and environmental opportunities.”

p. 7

“We employed the widely used COM-B model [18] which is linked to intervention frameworks, such as the Behaviour Change Wheel (BCW), and can, therefore, derive information about the most suitable interventions to improve corresponding behaviour.”

p. 32

“In this study, we applied a systematic framework – the COM-B model [18] – to explore the previous barriers and facilitators to consideration of using PrEP, use, as well as overarching factors constituting barriers and facilitators to HIV prevention more broadly data among people now living with HIV.”

---

R2: Some details regarding the qualitative methods are missing. Could you please address these in the manuscript?

Since the reviewer did not specify which details are missing, we used the Standards for Reporting Qualitative Research (SRQR) checklist – Methods section – to compare our reporting against:

• Qualitative approach and research paradigm: We clarified more strongly our qualitative approach. We had already addressed the paradigm that we pursued:

p.7

“We conducted a thematical framework analysis of semi-structured interview data. […] We therefore take a pragmatist approach to the data [25]. “

• Researcher characteristics and reflexivity: We had already discussed researcher characteristics such as professional background and experience as well as whether the interviewers had previous contact with the interviewees. However, we have added more specific background information on the interviewers and more clearly described the authors’ qualifications:

p. 11

“Two white British women (of whom HC is a co-author) with experience in qualitative interviewing and with a professional background in sexual health contacted the participants to arrange the interviews and provide an opportunity to ask questions prior to the interview.”

pp. 14, 15

“The study team drew from a variety of disciplines and methodological backgrounds. CHö and DM worked as Behavioural Scientists (CHö has a PhD in Social Psychology, DM has a Master’s degree in Osteopathy ), HK, AS and AA as HIV/ STI Surveillance and Prevention Scientists (HK and AS have a Master’s degree in the Control of Infectious Diseases, and AA has a PhD in surveillance and epidemiology of HIV), HC was a Sexual Health and HIV Lead and holds a Bachelor degree in Social Anthropology and a Masters in Global Public Health, and TD (FFPH PhD) and CHu a Consultant Epidemiologist (FFPH, MSc Demography and Health).”

p. 15

“The two interviewers had previously worked as sexual health advisors and were familiar with working on HIV and did not hold stigmatising attitudes.”

• Context: The possible influence of wider context information (the PrEP Impact Trial and Covid-19) had already been provided. We added a sentence on the immediate physical context of participants:

p. 12

“Most participants were by themselves somewhere private when they took part which allowed talking about sensitive topics.”

pp. 13, 14

“We acknowledge that background factors may have influenced the data. Although none of our participants took part in the PrEP Impact Trial [3], they were recruited during or after its conclusion, so the identified barriers and facilitators for PrEP use vary due to the change in PrEP availability and policies. The interviews also overlapped with the COVID19 pandemic. Access to sexual healthcare and consultation were impacted by this, which may have further negatively influenced the uptake of PrEP [33]. Throughout the analysis, we provide the diagnosis date, where relevant, to provide context on these potential influences.”

• Sampling strategy: We had already addressed how and why participants were selected for this study, but we now added more information on why the recruitment was halted and provided a reflective discussion on the issue of data saturation:

p. 12

“Halting recruitment was consequently determined by logistical reasons rather than due to data saturation.”

p. 14

“At the time when the current authors took over from the research team who had initiated the interview study, the final sample was predetermined (see Data collection for the reasons on that). Therefore, we cannot conclusively determine whether thematic saturation has been achieved. How

---

## [Decision Letter · Decision Letter 1]

23 Dec 2025

Retrospective challenges to pre-exposure prophylaxis (PrEP) use among people living with HIV – A qualitative analysis using the COM-B framework.

PONE-D-25-26292R1

Dear Dr. Hoerst,

We’re pleased to inform you that your manuscript has been judged scientifically suitable for publication and will be formally accepted for publication once it meets all outstanding technical requirements.

Kind regards,

Douglas S. Krakower, MD

Academic Editor

PLOS One

Additional Editor Comments (optional):

Reviewers' comments:

Reviewer's Responses to Questions

**Comments to the Author**

1. If the authors have adequately addressed your comments raised in a previous round of review and you feel that this manuscript is now acceptable for publication, you may indicate that here to bypass the “Comments to the Author” section, enter your conflict of interest statement in the “Confidential to Editor” section, and submit your "Accept" recommendation.

Reviewer #1: All comments have been addressed

2. Is the manuscript technically sound, and do the data support the conclusions?

Reviewer #1: Yes

3. Has the statistical analysis been performed appropriately and rigorously? 

Reviewer #1: Yes

4. Have the authors made all data underlying the findings in their manuscript fully available?

Reviewer #1: Yes

5. Is the manuscript presented in an intelligible fashion and written in standard English?

Reviewer #1: Yes

6. Review Comments to the Author

Reviewer #1: (No Response)

7. PLOS authors have the option to publish the peer review history of their article (what does this mean? ). If published, this will include your full peer review and any attached files.

**Do you want your identity to be public for this peer review?** For information about this choice, including consent withdrawal, please see our Privacy Policy .

Reviewer #1: No

---

## [Editor Report · Acceptance letter]

PONE-D-25-26292R1

PLOS One

Dear Dr. Hörst,

I'm pleased to inform you that your manuscript has been deemed suitable for publication in PLOS One. Congratulations! Your manuscript is now being handed over to our production team.

Kind regards,

on behalf of

Dr. Douglas S. Krakower

Academic Editor

PLOS One